

# The reading difficulties in Chinese for individuals with attention deficit hyperactivity disorder: the role of executive function deficits

Jia Wei[1], Dengxian Yang[1], Fang Cheng[1,2] and Wenwu Zhang[1,2]

[1] Ningbo University Health Science Center, NingBo, China
[2] Department of Psychiatry, Affiliated Kangning Hospital of Ningbo University, NingBo, China

## ABSTRACT

**Background:** Attention-deficit/hyperactivity disorder (ADHD) and reading disability (RD) are prevalent childhood conditions that together affect millions of children worldwide. In China, the prevalence of ADHD is approximately 6.4%, whereas the prevalence of RD ranges from 3.45% to 8%. Approximately 18–45% of children with ADHD also display comorbid RD, which further compromises their academic success and social functioning. Executive-function (EF) deficits are a core feature of ADHD and are known to affect reading in RD, yet their specific impact on Chinese reading remains under-explored.

**Objective:** This study investigated how EF deficits influence Chinese reading in children with ADHD, with the goal of informing diagnosis and intervention for ADHD-RD comorbidity.

**Methods:** This study recruited 160 Chinese-speaking children who met DSM-5 criteria for ADHD and allocated them to two groups—ADHD-only ($n$ = 80) and ADHD + RD ($n$ = 80). ADHD symptoms were rated with the Swanson, Nolan and Pelham Teacher and Parent Rating Scale (SNAP-IV), whereas Chinese reading was evaluated with the Dyslexia Checklist for Chinese Children (DCCC). Executive functions were measured with tasks tapping visuospatial working memory, verbal working memory, and response inhibition (Go/No-Go).

**Results:** Both groups showed no significant differences in ADHD symptom scores. Compared with the ADHD-only group, the ADHD + RD group obtained higher total and subscale DCCC scores and lower accuracies on EF tasks. Total DCCC scores correlated negatively with EF performance, especially on visuospatial working-memory and response-inhibition tasks.

**Conclusion:** This study suggests that individuals with ADHD comorbid with Chinese reading disabilities (RD) exhibit more pronounced deficits in executive function, particularly in verbal and visual-spatial working memory, and response inhibition tasks, compared to individuals with ADHD alone. These cognitive deficits are significantly negatively correlated with Chinese reading abilities, emphasizing the importance of not only focusing on traditional ADHD symptoms but also prioritizing training to enhance executive functions, especially visual-spatial working memory and response inhibition, when diagnosing and treating patients with ADHD comorbid with RD, in order to improve their reading abilities.

Corresponding authors
Fang Cheng,
chengfang198220@sina.com
Wenwu Zhang, knyyzww@163.com

# INTRODUCTION

Attention-deficit/hyperactivity disorder (ADHD) is a common neurodevelopmental condition that affects children worldwide during their formative years. According to global health data, the average prevalence of ADHD among children and adolescents is approximately 5% to 7%, while in school-aged children in China, this proportion reaches around 6.4% (*Polanczyk et al., 2014*). Reading disabilities (RD) represent another common obstacle to learning, particularly prevalent in regions where Chinese is the native language, with a prevalence rate ranging from 3.45% to 8% (*Zhang et al., 1996*; *Germanò, Gagliano & Curatolo, 2010*). Despite having access to education, normal cognitive function, and intelligence, children with reading disabilities struggle significantly with literacy. These figures underscore the burden on families and the education system.

Comorbidity estimates suggest that 18–45% of children with ADHD also meet criteria for RD (*DuPaul, Gormley & Laracy, 2013*). This heightened rate of comorbidity exacerbates the risks of academic difficulties, psychosocial consequences, and long-term adverse effects, which may persist into adulthood (*Moura et al., 2017*). On the other hand, children with ADHD most commonly struggle with attentional issues, often accompanied by hyperactivity and/or impulsivity, which frequently lead to social difficulties and poor academic performance. Similar to the etiology of RD, the precise nature and etiology of ADHD remain subjects of debate (*Willcutt et al., 2005a*; *Serrallach et al., 2016*). In terms of academic performance, high-motivation ADHD children perform similarly to control groups, with academic issues being attributed to behavioral rather than cognitive problems. Specifically, hyperactivity and impulsivity have been shown to negatively impact academic achievement (*Geurts & Embrechts, 2008*; *Staikova et al., 2013*). Pointing out the issue of comorbidity, children with comorbid reading, mathematical, or spelling deficits along with attention deficits may suffer greater impairments in learning compared to children with ADHD alone (*Gut et al., 2012*).

Executive functions (EF) refer to higher-order processes—such as working memory, response inhibition, and cognitive flexibility—that enable goal-directed behaviour. In the acquisition and development of reading skills, executive function plays a central role (*Germanò, Gagliano & Curatolo, 2010*; *Doebel, 2020*). Executive function deficits are not only fundamental neurocognitive characteristics of ADHD (*Barkley, 1997*) but are also associated with manifestations of reading disabilities, such as difficulties in text comprehension and word recognition processes (*Swanson et al., 2003*). Research indicates that in ADHD patients comorbid with reading disabilities, the deficits in executive function exhibit more complex and unique manifestations (*August & Garfinkel, 1990*). These patients not only perform poorly on traditional executive function tasks but also face greater challenges in specific language-related cognitive tasks (*Willcutt et al., 2005b*; *Pennington, 2006*). Phenotypic analysis further reveals a broad overlap of cognitive deficits

between ADHD and RD, rather than singular primary cognitive deficits (*McGrath et al., 2011*).

However, little is known about how EF deficits hinder Chinese reading in children with ADHD. Most existing studies use English or other alphabetic scripts; few examine EF in the context of RD within a logographic language such as Chinese (*Serrallach et al., 2016*). Cross-script perspective on reading disorders. Unlike alphabetic scripts, which map graphemes onto phonemes in a relatively linear fashion, modern Chinese is a morpho-syllabic writing system: each character encodes a syllable–morpheme pair and conveys meaning through semantic radicals while signalling pronunciation only indirectly through phonetic radicals (*Perfetti & Tan, 1998*; *Shu & Anderson, 1997*). This logographic structure results in greater visuospatial and orthographic-analysis demands (*e.g.*, stroke configuration, radical position) and weaker grapheme-to-phoneme transparency compared with shallow alphabetic languages such as Finnish or Spanish and even deep alphabetic scripts such as English (*Ziegler & Goswami, 2005*). Consequently, Chinese reading difficulties often manifest as deficits in visual-spatial working memory, semantic-radical awareness and rapid visual discrimination, whereas alphabetic dyslexia is more strongly linked to phonological decoding and letter-sound integration (*McBride, 2016*; *Chung & Lam, 2020*). Because these skills fall under the umbrella of executive functions—particularly visuospatial working memory and inhibitory control—examining EF deficits in Chinese-speaking children with ADHD provides a unique opportunity to clarify whether the cognitive profile of ADHD-related reading problems is script-specific or cross-linguistically stable.

The unique characteristics of the Chinese language structure and reading process, such as the processing of complex characters and the association between sound and meaning, pose specific demands on executive function (*Le Cunff, Dommett & Giampietro, 2024*). Moreover, current research often fails to fully explore the unique patterns of executive function in comorbid ADHD and RD within a Chinese context (*Baddeley, Gathercole & Papagno, 1998*; *Wen, Biedroń & Skehan, 2016*). Many studies lack in-depth analysis contrasting different cognitive deficits and their relationship with Chinese reading disabilities, as well as fail to specify the specific impacts of these deficits on daily functioning and academic achievement.

To address this gap, we conducted a systematic study of 160 Chinese-speaking children diagnosed with ADHD under DSM-5 criteria. It will analyze the executive function characteristics of 80 individuals with ADHD alone and 80 individuals with comorbid ADHD and RD. ADHD symptoms will be assessed using the SNAP-IV scale, while reading abilities will be evaluated using the dyslexia checklist for Chinese children (DCCC). Additionally, executive function will be thoroughly measured through tasks involving visual-spatial working memory, verbal working memory, and response inhibition (Go/No-go task). The primary objective of this study is to elucidate the unique characteristics of executive function in comorbid ADHD and RD populations, providing scientific evidence for future personalized treatment plans and intervention strategies.

## METHODS

### Ethics statement

The study protocol was reviewed and approved by the Ethics Committee of Ningbo Kangning Hospital (approval No. NBKNYY-2020-LC-50, 12 August 2021). All procedures conformed to the Declaration of Helsinki. Parents or legal guardians provided written informed consent after receiving a detailed description of study aims, potential risks (*e.g.*, possible emotional distress when answering reading-difficulty items), and confidentiality limits, including the hospital's mandatory-reporting policy. Child participants gave written or oral assent in age-appropriate language. All questionnaires were completed anonymously; datasets were de-identified, stored on an encrypted drive, and shared only in aggregate form.

### Study design

We employed a cross-sectional design to compare executive-function (EF) profiles and Chinese reading abilities between two subgroups of children with ADHD: (i) ADHD-only and (ii) ADHD + RD. Data collection was carried out from 1 September 2021 to 30 September 2023. Figure 1 presents the recruitment and allocation flow.

### Participants

A total of 212 consecutive patients aged 6–16 years who attended the Children and Adolescent Psychiatry Clinic, Ningbo Kangning Hospital, were screened. Inclusion criteria were: (1) Diagnosis of ADHD confirmed by two licensed child psychiatrists according to DSM-5; (2) Full-scale IQ ≥ 70 (WISC-IV Chinese version); (3) Normal or corrected-to-normal vision and hearing; (4) No ADHD- or RD-related medication or behavioural intervention in the previous 3 months.

Exclusion criteria comprised: autism spectrum disorder, other major neurodevelopmental disorders, psychotic disorders, neurological disease, or uncorrected sensory deficits. Thirty-two children were excluded (17 did not meet IQ criterion; 15 declined participation). The remaining 160 participants were stratified by their Dyslexia Checklist for Chinese Children (DCCC) total score (cut-off ≥ −1 SD) into an ADHD-only group ($n = 80$) and an ADHD + RD group ($n = 80$).

### Survey content

In order to comprehensively assess the executive function and reading ability of ADHD patients, this study utilized the following measurement tools:

#### Wechsler intelligence scale for children

Wechsler intelligence scale for children (WISC) is a widely used intelligence assessment tool designed for children and adolescents aged 6 to 16 years and 11 months. It comprises various subtests, such as vocabulary, reasoning, digit span, and spatial abilities, to comprehensively evaluate children's performance across different domains of intelligence. The scores on this scale are based on a standard intelligence quotient (IQ) system with a mean of 100 and a standard deviation of 15 (*Zhang, 2009*). All patients included in this
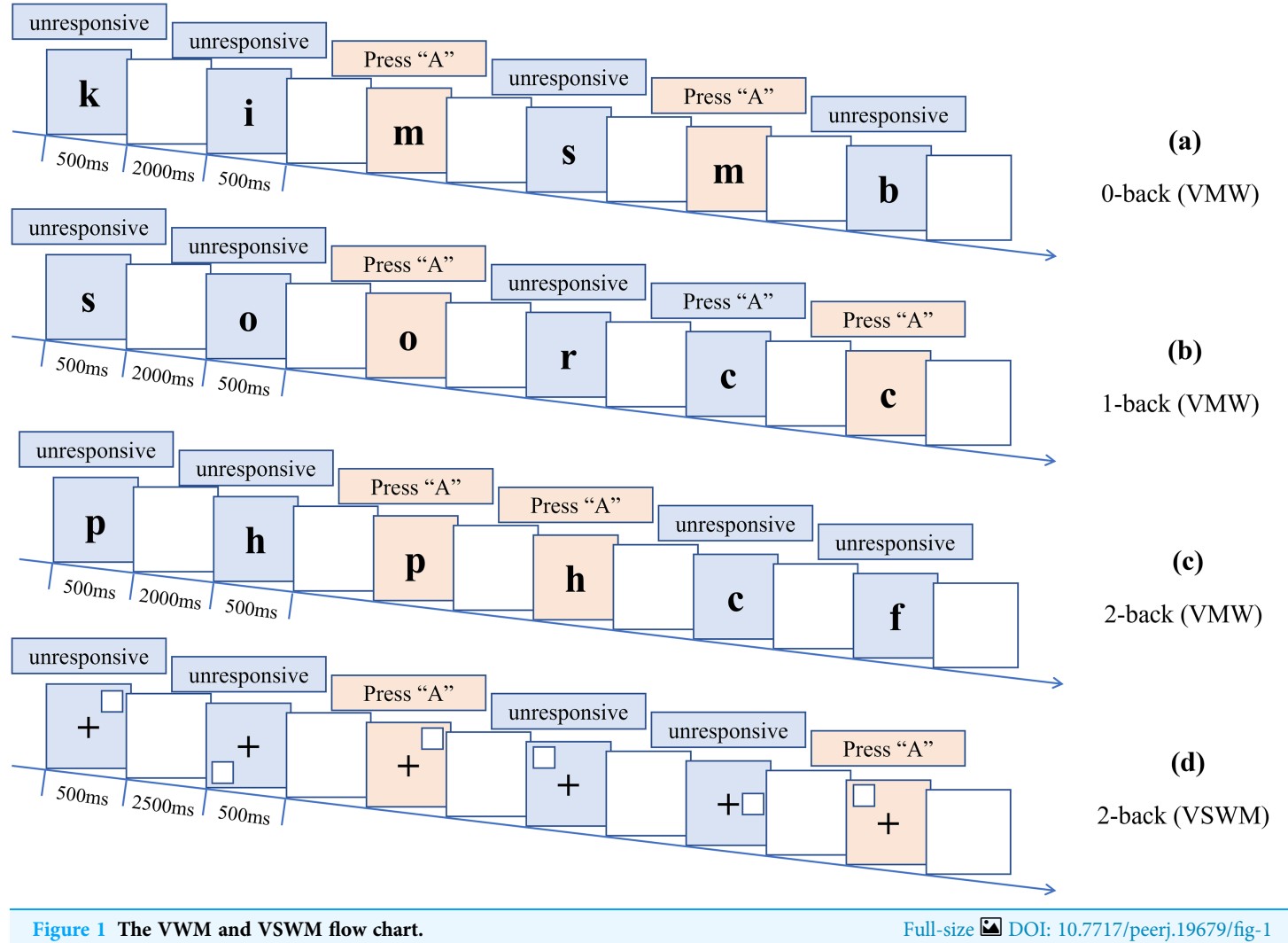

**Figure 1 The VWM and VSWM flow chart.**

study exhibited a general intelligence level, with overall intelligence quotient scores reaching within the average range.

### Disruptive behavior disorder rating scale

This scale consists of 18 items assessing ADHD symptoms and eight items assessing oppositional defiant symptoms, with each item rated on a four-point scale ranging from 0 to 3. The scale is designed for parent rating to assess the severity of behavioral disorders in children, with higher scores indicating more severe symptoms. It has demonstrated good reliability and validity (*Su et al., 2006*).

### Dyslexia checklist for Chinese children

The DCCC (*Wu, Song & Yao, 2006*) is a parent-rated checklist developed specifically for identifying Chinese-language reading difficulties. It comprises 37 items spanning eight skill domains: visual perception, auditory perception, semantic comprehension, attention control, rapid naming, fine-motor/handwriting, learning habits, and emotional–behavioural

adjustment. Each item is scored on a five-point Likert scale (1 = never, 5 = always); higher totals indicate more severe reading problems. Example items include: "Often skips or omits characters while reading". "Has trouble distinguishing visually similar characters such as '晴' *vs*. '睛'.

Large-sample validation shows excellent internal consistency (Cronbach's α = 0.97 for the total scale; α = 0.75–0.90 for subscales) and satisfactory 2-week test–retest reliability (r = 0.73). Consistent with *Hou et al. (2018)*, we classified a child as having RD when the age-adjusted total score lay ≥1 SD above the normative mean; scores below this cut-off were treated as typical reading attainment. The DCCC has demonstrated strong convergence with timed Chinese character-recognition tests, supporting its validity as a screening tool even though it is completed by parents.

### Verbal working memory and visual spatial working memory tasks

The experiments were designed using the Inquisit software, which included tasks for verbal working memory (VWM) and visual spatial working memory (VSWM), employing the N-back task paradigm. The tasks involved presenting letters or positions on the computer screen and requiring patients to recall and identify specific sequences.

The VWM task, as shown in Figs. 1A–1C, is divided into three blocks. The first block is the 0-back block, consisting of 30 trials. When the letter "m" appears on the screen, the patient is required to quickly press the "A" key, while ignoring other letters. The second block is the 1-back block, comprising 31 trials. When the current letter on the screen is the same as the letter from the previous trial, the patient needs to quickly press the "A" key. The third block is the 2-back block, consisting of 32 trials. When the current letter on the screen is the same as the letter from two trials ago, the patient needs to quickly press the "A" key. Each letter is presented for 500 ms with an interstimulus interval of 2,000 ms. Each block is practiced 10 times for a total of 30 times, ensuring that the patient fully understands the task instructions.

The VSWM task, as shown in Fig. 1D, is conducted after completing the VWM task. In this task, only the 2-back block is performed, comprising 42 trials. A fixation point "+" is displayed at the center of the screen, and a blue square randomly appears in one of the eight positions within a (3 × 3) grid (excluding the center point). When the position of the blue square on the current trial matches the position of the blue square from two trials ago, the patient is instructed to press the "A" key. Each blue square is presented for 500 ms with an interstimulus interval of 2,500 ms. The 2-back block is practiced 12 times, ensuring that the patient fully understands the task instructions.

### Go/No-go task

The Go/No-go task, programmed using the Inquisit software, is designed to assess the patient's inhibitory control ability. In this task, participants are required to respond quickly to the appearance of a "mole" pattern while inhibiting their response when an "eggplant" pattern appears. Patients are instructed to respond to the "Go" trials, which consist of 14 variations of the "mole" pattern with slight differences in appearance, by quickly pressing the spacebar. Conversely, for the "No-go" trials, which feature the "eggplant" pattern,

patients must refrain from pressing the spacebar when the "eggplant" pattern appears. The task comprises a total of 165 trials, with 124 "Go" trials and 41 "No-go" trials. The interstimulus interval is set to 1,800 ms for "Go" trials and 1,300 ms for "No-go" trials, with their appearances following a pseudo-random sequence. This task is designed to reveal potential issues in attention control and response inhibition in ADHD patients.

## Data analysis methods

After logic checking and proofreading, we used SPSS 23.0 (IBM Corp., Armonk, NY, USA) to process and analyze the data. To comprehensively assess and interpret the research data, the following detailed data analysis methods were employed in this study:

Descriptive statistics: Initially, descriptive statistical analysis was conducted on baseline data for all participants, including intelligence, reading ability, ADHD symptoms, and executive function test results. This involved calculating the mean, standard deviation, minimum, and maximum values to provide an overview of the data distribution.

Independent samples t-test: The independent samples t-test was utilized to compare the differences between the ADHD-along group and the ADHD comorbid with RD group on measures of intelligence (Wechsler Intelligence Scale for Children), total scores, and subscale scores on the Parent-rated Disruptive Behavior Disorder Rating Scale (DBDRS). This analysis method was employed to assess whether there were significant differences in means between the two groups on continuous variables.

Pearson correlation analysis: Pearson correlation analysis was conducted to explore the associations between scores on the DCCC scale and scores on ADHD symptomatology, as well as the accuracy rates of various executive tasks. This analysis aids in identifying linear relationships between reading disability scores and other variables.

Analysis of covariance (ANCOVA): Utilizing ANCOVA, with the DCCC total score as the dependent variable, ADHD total symptom score as the fixed factor, while controlling for age, Wechsler Intelligence Scale for Children (WISC) score, accuracy rates on the VSWM task (2-back), VWM task (0-back, 1-back, 2-back), and response inhibition task. This method helps elucidate the independent effects of single or multiple predictors on the variability of the dependent variable while controlling for other variables.

Significance testing: All statistical tests were set at a significance level of $\alpha = 0.05$. Appropriate F-statistics and $p$-values were utilized to report the significance of each factor in the analysis of covariance, ensuring the statistical validity of the results.

Through the comprehensive statistical methods described above, this study aimed to accurately assess the relationship between ADHD and reading disabilities, and delve into how executive function deficits impact the reading ability of ADHD patients. The selection of these methods ensured the scientific rigor of the study results, providing reliable data support for future intervention strategies and treatment recommendations.

## RESULTS

### Univariate between-group comparative analysis

In this study, we compared the demographic characteristics and various assessment indicators between the ADHD-along group (80 cases) and the ADHD comorbid with

reading disabilities (ADHD+RD) group (80 cases), as shown in Table 1. No statistically significant differences were observed between the two groups in terms of gender (81.25% male) and age (mean 9.92 years, standard deviation approximately 1.02). (gender: $\chi^2 = 0.00$, $p = 1.000$; age: t = −0.08, $p = 0.938$). The comparison of SNAP-IV total scores and its subscales (inattention, hyperactivity/impulsivity, oppositional symptoms) also showed no significant differences between the two groups. For instance, the total scores were 36.16 ± 12.41 for the ADHD-along group and 37.20 ± 13.08 for the ADHD+RD group (t = −0.52, $p = 0.607$).

The ADHD+RD group scored significantly higher than the ADHD-along group on both the total score and all subscales of the dyslexia checklist for Chinese children (DCCC). For instance, there was a significant difference in the total score of the Children's Chinese Dyslexia Checklist (DCCC) between the pure ADHD group (145.81 ± 36.77) and the ADHD+RD group (177.13 ± 31.62) (t = −5.78, $p < 0.001$). In the executive function tests, the ADHD+RD group exhibited significantly lower accuracy rates in the VWM task (*e.g.*, 1-back task accuracy: pure ADHD group 0.96 ± 0.05 compared to ADHD+RD group 0.88 ± 0.11, t = 6.17, $p < 0.001$) and the VSWM task (*e.g.*, accuracy of ADHD-along group 0.88 ± 0.04 compared to ADHD+RD group 0.73 ± 0.09, t = 12.77, $p < 0.001$), as well as the Go/No-go task (ADHD-along group 0.97 ± 0.02 compared to ADHD+RD group 0.94 ± 0.04, t = 7.13, $p < 0.001$), compared to the pure ADHD group.

These findings reveal significant differences in the severity of reading disabilities and executive function performance among children with ADHD comorbid with reading disabilities, suggesting that this group faces greater challenges in cognition and executive functioning.

## Correlation analysis results

In analyzing the association between ADHD symptoms, executive function performance, and the severity of reading disabilities, we observed some interesting patterns. As shown in Table 2, the correlation results between DCCC scores, ADHD symptoms, and performance on executive function tasks demonstrate significant associations among these variables.

### *Correlation between DCCC scores and ADHD symptoms*
The total score of DCCC and some of its subscales show a significant positive correlation with the total score of SNAP-IV. Specifically, the correlation coefficient between the total score of DCCC and SNAP-IV is 0.225 ($p < 0.01$), indicating a positive correlation between the severity of reading disabilities and the overall manifestation of ADHD symptoms. Additionally, the total score of DCCC shows a statistically significant positive correlation with the scores of inattention (IA) and hyperactivity/impulsivity (IH) from SNAP-IV. Specifically, the correlation coefficient between the total score of DCCC and IA score is 0.222 ($p < 0.01$), and with IH score is 0.198 ($p < 0.05$). This indicates that the more severe the symptoms of inattention and hyperactivity/impulsivity are, the more severe the reading disabilities tend to be.

**Table 1 Comparison of general information and various assessments between the two groups.**

| | | Group (M ± SD) | | | |
| | | ADHD group (n = 80) | ADHD+RD group (n = 80) | $t/\chi^2$ | $p$ |
|---|---|---|---|---|---|
| Gender | Male | 65 (81.25%) | 65 (81.25%) | 0.00 | 1.000 |
| | Female | 15 (18.75%) | 15 (18.75%) | | |
| Age | | 9.91 ± 1.02 | 9.93 ± 1.02 | −0.08 | 0.938 |
| SNAP-IV | Total | 36.16 ± 12.41 | 37.20 ± 13.08 | −0.52 | 0.607 |
| | IA | 15.95 ± 4.36 | 15.86 ± 4.69 | 0.12 | 0.903 |
| | IH | 11.21 ± 5.35 | 12.00 ± 6.10 | −0.87 | 0.387 |
| | ODD | 9.00 ± 5.44 | 9.34 ± 4.42 | −0.43 | 0.667 |
| WISC | | 103.33 ± 7.71 | 103.05 ± 7.77 | 0.23 | 0.822 |
| DCCC | Total | 145.81 ± 36.77 | 177.13 ± 31.62 | −5.78 | 0.000*** |
| | DCCC[1] | 16.79 ± 5.04 | 22.26 ± 5.49 | −6.57 | 0.000*** |
| | DCCC[2] | 15.24 ± 4.65 | 18.80 ± 4.89 | −4.72 | 0.000*** |
| | DCCC[3] | 23.69 ± 7.54 | 29.30 ± 5.82 | −5.27 | 0.000*** |
| | DCCC[4] | 21.56 ± 6.29 | 25.50 ± 5.75 | −4.13 | 0.000*** |
| | DCCC[5] | 14.74 ± 5.50 | 18.23 ± 4.57 | −4.36 | 0.000*** |
| | DCCC[6] | 20.68 ± 6.46 | 24.63 ± 4.40 | −4.52 | 0.000*** |
| | DCCC[7] | 14.71 ± 4.79 | 17.84 ± 4.13 | −4.42 | 0.000*** |
| | DCCC[8] | 18.41 ± 4.33 | 20.58 ± 4.19 | −3.21 | 0.002** |
| VWM | 0-back | 0.99 ± 0.02 | 0.97 ± 0.04 | 2.40 | 0.018* |
| | 1-back | 0.96 ± 0.05 | 0.88 ± 0.11 | 6.17 | 0.000*** |
| | 2-back | 0.86 ± 0.06 | 0.80 ± 0.09 | 5.06 | 0.000*** |
| VSWM | | 0.88 ± 0.04 | 0.73 ± 0.09 | 12.77 | 0.000*** |
| Go/No-go | | 0.97 ± 0.02 | 0.94 ± 0.04 | 7.13 | 0.000*** |

**Notes:**
* $p$ <0.05.
** $p$ <0.01.
*** $p$ <0.001.
DCCC[1] = visual perception disorder; DCCC[2] = auditory perception disorder; DCCC[3] = meaning comprehension disorder; DCCC[4] = writing disorder; DCCC[5] = oral language disorder; DCCC[6] = written expression disorder; DCCC[7] = poor reading habits disorder; DCCC[8] = inattention disorder.

### Correlation between DCCC scores and executive function task performance

The performance on the VWM and VSWM tasks showed a significant negative correlation with the total score and sub-items of DCCC. In particular, there is a significant negative correlation between the total score of DCCC and the performance in VSWM (r = −0.332, $p$ < 0.01), indicating that poorer spatial working memory is associated with more severe reading disabilities. Similarly, the total score of DCCC also shows a significant negative correlation with the performance in the 1-back task of VWM (r = −0.247, $p$ < 0.01).

In a more detailed analysis of DCCC sub-items, the negative correlation between visual perceptual difficulties (DCCC1) and VSWM performance is particularly pronounced (r = −0.420, $p$ < 0.01), suggesting a close association between the decline in visual spatial working memory capacity and visual perceptual difficulties. Furthermore, in the verbal working memory task, the visual perceptual difficulties (DCCC1) of DCCC also exhibit a

**Table 2 Correlation analysis of DCCC scores with ADHD symptoms and EF task performance.**

| | DCCC total | DCCC[1] | DCCC[2] | DCCC[3] | DCCC[4] | DCCC[5] | DCCC[6] | DCCC[7] | DCCC[8] |
|---|---|---|---|---|---|---|---|---|---|
| VWM (0-back) | −0.132 | −0.228** | −0.144 | −0.073 | −0.157* | −0.005 | −0.162* | −0.111 | 0.033 |
| VWM (1-back) | −0.247** | −0.364** | −0.197* | −0.183* | −0.184* | −0.212** | −0.213** | −0.178* | −0.093 |
| VWM (2-back) | −0.103 | −0.256** | −0.134 | −0.061 | −0.021 | −0.025 | −0.076 | −0.155* | 0.045 |
| VSWM | −0.332** | −0.420** | −0.243** | −0.308** | −0.220** | −0.227** | −0.304** | −0.295** | −0.173* |
| Go/No-go | −0.330** | −0.333** | −0.275** | −0.293** | −0.216** | −0.318** | −0.274** | −0.295** | −0.191* |
| SNAP-IV total | 0.225** | 0.134 | 0.314** | 0.190* | 0.187* | 0.123 | 0.163* | 0.186* | 0.237** |
| IA | 0.222** | 0.183* | 0.324** | 0.179* | 0.178* | 0.147 | 0.153 | 0.087 | 0.254** |
| IH | 0.198* | 0.092 | 0.277** | 0.170* | 0.144 | 0.106 | 0.147 | 0.199* | 0.224** |
| ODD | 0.147 | 0.071 | 0.191* | 0.127 | 0.151 | 0.06 | 0.108 | 0.169* | 0.118 |

Notes:
* $p < 0.05$.
** $p < 0.01$.

DCCC[1] = visual perception disorder; DCCC[2] = auditory perception disorder; DCCC[3] = meaning comprehension disorder; DCCC[4] = writing disorder; DCCC[5] = oral language disorder; DCCC[6] = written expression disorder; DCCC[7] = poor reading habits disorder; DCCC[8] = inattention disorder.

significant negative correlation with the 1-back task performance of VWM ($r = −0.364$, $p < 0.01$).

The results of the Go/No-go task also show a significant negative correlation with various sub-items of DCCC, such as the correlation coefficient of DCCC total score with Go/No-go task performance being −0.330 ($p < 0.01$). This indicates that children who perform poorly in inhibitory control tasks often exhibit more severe difficulties in various aspects of reading disorders.

These correlation analysis results not only reveal the connection between ADHD symptoms and reading disorders but also underscore the significance of executive function deficits in reading disorders. Specifically, there is a significant correlation between deficits in attention and working memory and the decline in reading ability. ANCOVA analysis of predictors of DCCC score.

When conducting the analysis of covariance (ANCOVA) to explore the potential predictive factors influencing DCCC scores, we considered multiple covariates along with a fixed factor. As shown in Table 3, the analysis results revealed the following key findings:

### The impact of covariates

The impact of covariates was considered, including age, WISC score, and performance on executive function tasks (VWM 0-back, 1-back, 2-back, VSWM, Go/No-go), on DCCC scores. Most variables did not show a significant effect on DCCC scores statistically. Specifically, age ($F = 0.33$, $p = 0.565$) and WISC score ($F = 0.13$, $p = 0.719$) did not significantly affect DCCC scores. Similarly, the results of the VWM tasks, including 0-back ($F = 0.00$, $p = 0.954$), 1-back ($F = 1.45$, $p = 0.231$), and 2-back ($F = 1.48$, $p = 0.226$) tasks, did not reach statistical significance.

However, the VSWM task ($F = 4.73$, $p = 0.032$) and the response inhibition (Go/No-go) task ($F = 4.56$, $p = 0.035$) demonstrated significant effects on DCCC scores. This suggests a significant correlation between the decline in visual spatial working memory capacity and

**Table 3 Predictors of DCCC score.**

|  | Type III sum of squares | df | F | p |
|---|---|---|---|---|
| Covariate |  |  |  |  |
| Age | 355.66 | 1 | 0.33 | 0.565 |
| WISC | 139.60 | 1 | 0.13 | 0.719 |
| VWM (0-back) | 3.51 | 1 | 0.00 | 0.954 |
| VWM (1-back) | 1,550.99 | 1 | 1.45 | 0.231 |
| VWM (2-back) | 1,584.79 | 1 | 1.48 | 0.226 |
| VSWM | 5,049.74 | 1 | 4.73 | 0.032* |
| Go/No-go | 4,872.37 | 1 | 4.56 | 0.035* |
| Fixed factor |  |  |  |  |
| SNAP-IV total | 78,782.81 | 52 | 1.42 | 0.068 |

**Note:**
  * $p < 0.05$.

response inhibition ability in specific domains of executive function and the severity of reading difficulties.

### Effect of fixed factors

The SNAP-IV total score, as a fixed factor, approached statistical significance in its impact on DCCC scores (F = 1.42, $p = 0.068$). This suggests a potential correlation between the overall severity of ADHD symptoms and the severity of reading difficulties, although further research is needed to confirm this result.

These ANCOVA analysis results allow us to identify the key executive function tasks associated with the severity of reading difficulties in children with ADHD. Specifically, visual spatial working memory and inhibitory control abilities significantly predict the severity of reading difficulties.

## DISCUSSION

### Theoretical implications of the findings

To our knowledge, this is the first study to systematically examine how distinct executive-function (EF) components relate to Chinese reading difficulty in children with ADHD. We found that the ADHD+RD group performed significantly worse than the ADHD-only group on VWM, VSWM and response-inhibition tasks, mirroring the pattern reported by *Gooch, Snowling & Hulme (2011)* in English readers. Unlike alphabetic scripts, Chinese is morpho-syllabic: characters integrate complex stroke configurations, semantic radicals and only indirect phonological cues (*Perfetti & Tan, 1998*; *McBride, 2016*). Reading therefore imposes heavier VSWM and visual-analysis demands and weaker grapheme–phoneme support than English or Spanish (*Chung & Lam, 2020*). Our data confirm this: deficits in VSWM showed the strongest correlation with DCCC scores (r = −0.42, $p < 0.01$), highlighting script-specific EF bottlenecks in ADHD-related reading problems. The findings fit a multiple-deficit account (*Willcutt et al., 2010*): ADHD is genetically linked to inhibition deficits, RD to working-memory deficits; their co-occurrence produces compounded, not merely additive, impairments.

Correlational analyses confirmed that poorer EF scores—especially on the visuospatial task—were moderately associated with greater Chinese reading difficulty. Two script-specific factors may explain why VSWM is particularly salient. First, Chinese characters require holistic visual analysis of complex stroke configurations, placing a higher load on spatial memory than alphabetic decoding (*Perfetti & Tan, 1998*). Second, phoneme-level mapping is absent; therefore, children must inhibit competing character candidates during recognition, aligning with the observed inhibition deficit.

Genetically, ADHD has been linked to response-inhibition pathways, whereas RD is associated with working-memory loci (*Willcutt et al., 2010*). Our findings suggest an interactive amplification rather than a linear addition of these vulnerabilities, consistent with twin data showing that the shared variance between reading difficulty and attention problems is largely attributable to the attentional component (*Greven et al., 2011*). Preliminary neuroimaging work suggests overlap in frontostriatal circuits (*e.g.*, right caudate volume; *McGrath & Stoodley, 2019*), offering a plausible neural basis that warrants direct testing in future multimodal studies.

## Practical implications

Given the strong EF–reading links, intervention should move beyond core ADHD symptoms to include EF-focused training. Randomised trials show that a 6-week visuospatial n-back protocol (20 min, three times per week) improves both VSWM and Chinese character recognition (*Lee, 2024*). Similarly, stop-signal or Go/No-go gamified drills can enhance response inhibition and sustain attention. In classrooms, visual chunking of stroke patterns and explicit teaching of semantic radicals may reduce VSWM load and accelerate character acquisition.

Furthermore, our findings underscore the importance of individualized interventions for children with ADHD comorbid with RD in clinical practice. For instance, training aimed at enhancing visual spatial working memory and response inhibition abilities may help improve reading skills and attentional control in these children. This is also consistent with the findings of *Lee (2024)*, who observed significant deficits in visual spatial working memory among Chinese children with reading difficulties. The stronger correlation between reading ability and attention deficit symptoms is supported by neuroimaging studies, which have found associations between ADHD and RD and the volume of frontal lobe regions, which are linked to attentional control (*Jagger-Rickels, Kibby & Constance, 2018*; *Kibby et al., 2020*, *2021*). Based on early developmental studies of large samples of twins, the genetic correlation between reading and attention deficit is significantly greater than that of hyperactivity, indicating that the substantial genetic overlap observed in the symptom domains of ADHD and reading difficulties is largely driven by attention deficits (*Greven et al., 2011*; *Paloyelis et al., 2010*).

## Strengths and limitations

This study is among the first to link distinct executive-function components to Chinese reading difficulty in children with ADHD, and it does so with a clinically diagnosed sample and well-validated EF tasks. Nevertheless, three limitations should be acknowledged. First,

the sample, though carefully screened, is modest in size and drawn from a single outpatient clinic, which may constrain statistical power and generalisability. Second, the cross-sectional design precludes conclusions about developmental trajectories; future longitudinal work could test how executive-function deficits and reading outcomes influence one another over time. Third, reading disability was identified with the parent-rated DCCC alone; although the scale shows strong psychometrics, incorporating an objective measure such as a timed character-recognition or oral-reading-fluency task would provide additional convergent validity. Addressing these issues in larger, multi-site, longitudinal studies—and examining how targeted interventions on visuospatial working memory or response inhibition alter reading progress—will be valuable next steps.

## CONCLUSION

This study has made a significant contribution to understanding the complex interaction between attention deficit hyperactivity disorder and reading disabilities in Chinese children. Our findings underscore the significant cognitive impairments present in children co-occurring with both ADHD and RD, particularly evident in tasks involving verbal working memory, visual spatial working memory, and response inhibition. These impairments are more pronounced compared to children with ADHD alone, indicating the development of a unique cognitive subtype with overlapping deficits that extend beyond the simple co-occurrence of ADHD and RD. The main conclusions include the following:

### Exacerbation of cognitive impairments
Children with comorbid ADHD and RD exhibit greater difficulties in executive functioning, which are closely linked to the severity of reading disabilities. This indicates that the co-occurrence of ADHD and RD creates a complex neurocognitive profile, necessitating targeted diagnostic and intervention strategies.

### Theoretical implications
The study findings support the hypothesis of shared neurobiological pathways or even genetic bases between ADHD and RD, contributing to the observed cognitive impairments. By confirming that these overlaps result in a unique pattern of cognitive deficits different from children with either condition alone, this study extends previous research.

### Clinical implications
The findings underscore the importance of developing targeted intervention strategies that address both the core symptoms of ADHD and specific executive function deficits associated with RD. Interventions focused on enhancing visual spatial working memory and improving response inhibition may be particularly beneficial in improving academic performance and daily functioning in children with comorbid ADHD and RD.

### Future directions

Despite providing valuable insights, this study is limited by its cross-sectional design and relatively small sample size. Future research should employ longitudinal studies to track cognitive deficits and their impact on ADHD and RD symptoms over time. Additionally, exploring the effectiveness of specific interventions targeting the unique cognitive deficits identified in comorbid ADHD and RD is crucial.

In conclusion, the interdependence between ADHD and RD necessitates consideration of the complex cognitive impairments present in affected children during diagnosis and treatment. As we continue to uncover the genetic and neurobiological basis of these conditions, our clinical strategies must evolve to effectively support this vulnerable population.

### Funding

This work was supported by the Zhejiang Province Natural Science Foundation of China (LGF19H090009), as well as the Medical and Health Technology of Zhejiang Province, China (2021KY330) and the Ningbo Natural Science Foundation, Zhejiang Province, China (202003N4262). The funders had no role in study design, data collection and analysis, decision to publish, or preparation of the manuscript.

### Grant Disclosures

The following grant information was disclosed by the authors:
Zhejiang Province Natural Science Foundation of China: LGF19H090009.
Medical and Health Technology of Zhejiang Province, China: 2021KY330.
Ningbo Natural Science Foundation, Zhejiang Province, China: 202003N4262.

### Competing Interests

The authors declare that they have no competing interests.

### Author Contributions

- Jia Wei conceived and designed the experiments, analyzed the data, prepared figures and/or tables, and approved the final draft.
- Dengxian Yang analyzed the data, prepared figures and/or tables, and approved the final draft.
- Fang Cheng performed the experiments, authored or reviewed drafts of the article, and approved the final draft.
- Wenwu Zhang conceived and designed the experiments, authored or reviewed drafts of the article, and approved the final draft.
## Human Ethics

The following information was supplied relating to ethical approvals (*i.e.*, approving body and any reference numbers):

The Ningbo Kangning Hospital granted Ethical approval to carry out the study within its facilities (Ethical Application Ref. No.: NBKNYY-2020-LC-50, 2020.10.15-2023.10.14).

## Data Availability

The raw data and codebook are available in the Supplemental Files.

## Supplemental Information

Supplemental information for this article can be found online at http://dx.doi.org/10.7717/peerj.19679#supplemental-information.

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
