# Peer review of "The reading difficulties in Chinese for individuals with attention deficit hyperactivity disorder: the role of executive function deficits"

_PeerJ, doi:10.7717/peerj.19679_

## Round 0.1 · original submission · Major Revisions

Dear Authors,

Thank you for submitting your manuscript to Peer J.
Please take into account the Reviewers' comments and suggestions and revise the paper accordingly.

Kind regards,
Marialaura Di Tella

Reviewer 1 ·

Basic reporting

While the writing is clear, there is room for improvement in sentence structure variation to enhance readability.
The current text does not mention figures or tables, nor whether raw data is publicly available. Including visuals and sharing raw datasets (e.g., via OSF or Figshare) would improve transparency and reproducibility.
The manuscript would benefit from a brief discussion comparing Chinese orthography with alphabetic systems in the context of reading disorders, which would help international readers better appreciate the study's relevance.

Experimental design

The article satisfactorily meets three out of the four evaluation criteria and demonstrates a clear contribution to the field, particularly in the study of executive functions within specific linguistic contexts such as Chinese. To achieve excellence across all categories, it is recommended that the authors include a more detailed description of the methods in the main body of the manuscript and clarify the ethical procedures followed.

Validity of the findings

The manuscript presents a meaningful contribution to the existing literature by exploring an under-researched area: the role of executive function deficits in reading difficulties among Chinese-speaking children with ADHD. While the impact and novelty are not explicitly emphasized, the study clearly addresses a relevant knowledge gap, and the rationale for the research is well stated. This work could serve as a valuable replication model in other linguistic contexts, provided that the authors frame its broader implications more explicitly.

Regarding the data quality, all underlying data appear to be methodologically robust and statistically sound.

The conclusions are clearly stated and remain tightly aligned with both the original research question and the results obtained. The authors avoid overgeneralization and focus on implications that are directly supported by the data, which strengthens the overall integrity of the study.

Additional comments

1. Broader theoretical contextualization:While the study appropriately focuses on the Chinese context, it could benefit from a broader discussion on how specific features of the Chinese language (such as logographic writing or the visual complexity of characters) may interact with executive function deficits. This would not only enrich the interpretation of the results but also add comparative value for readers from other linguistic backgrounds.
2. Further elaboration on clinical implications: Although general clinical recommendations are mentioned, it would be helpful for the authors to more explicitly detail how the findings could be applied in educational or therapeutic settings—for example, through targeted interventions aimed at executive functions such as visual-spatial working memory or response inhibition.

Annotated reviews are not available for download in order to protect the identity of reviewers who chose to remain anonymous.

Reviewer 2 ·

Basic reporting

1. On line 47, it says “this data underscores …”. Data is a plural name it should be these data underscore”.
2. Previous research into ADHD and executive functions (EF) is well documented. However, there is scarce evidence on reding difficulties (RD), particularly on difficulties when reading on Chinese. As most of the available research is focused on alphabetic reading systems, it would be interesting to describe at least the general particularities of learning to read in the Chinese writing system. This is necessary because it is the main argument for supporting the relation between EF and difficulties in reading in Chinese.

3. References should follow APA recommendations or at least be presented in alphabetic order and include DOI.

Experimental design

4. Given the wide age range, more detailed demographic data would be desirable. Note that the reading ability of 6-year-olds learning to read may not be easily comparable to that of 14-16-year-olds with years of reading experience.
5. A better methodology for diagnosing reading difficulties is missing. Scales are common tools for diagnosing ADHD. However, in the case of reading difficulties, scales are often used at a first screening phase to then perform a specific assessment. I am not familiarized with the Dyslexia Checklist for Chinese Children (DCCC), and I assume that many of the non-Chinese researchers must not be familiar with it either. It would be useful to at least present some samples of the main items of the scale mostly when the checklist is rated by parents. Subjectivity could be a strong limitation on the checklist reliability.
6. Some measures to assess reading would be convenient. That way, EF and reading abilities could be more reliably related.

Validity of the findings

7. It is not a new finding that individuals with double deficits present greater impairments than those with only one deficit What is interesting is to know what specific deficits result differential between the groups considered in this research.
The discussion should be more extended since the authors have findings that yield interesting information.
8. On the other hand, to support their results they resort to studies on gray matter that, although interesting, are not directly related to the present study. At the same time, comments on studies on reading deficit and ADHD comorbidities are scarce.

Additional comments

Introduction and Discussion should be improved in order to include futher evidence on Chinese reading difficulties, and ADHD and reading difficulties comorbidity.

Annotated reviews are not available for download in order to protect the identity of reviewers who chose to remain anonymous.

---

## Round 0.2 · accepted · Accept

Dear Authors,

Thank you for submitting your manuscript to Peer J.
The Reviewers' comments and suggestions have been adequately addressed. Therefore, the manuscript can be accepted for publication.

Kind regards,
Marialaura Di Tella

Reviewer 1 ·

Basic reporting

The article meets key scholarly criteria and contributes meaningfully to the understanding of neurodevelopmental disorders in a non-Western linguistic context.

Experimental design

The study presents original and well-structured research focused on a specific and relevant population. The research question is clear and addresses an important gap in knowledge regarding the relationship between ADHD, reading disability, and executive functions in Chinese-speaking children. The methodology is rigorous, using recognized diagnostic criteria and validated assessment tools. Although brief, the methods section provides enough detail to suggest the study is replicable. Overall, the work demonstrates a high technical and scientific standard, aligned with expected ethical and academic norms.

Validity of the findings

The study’s results are well presented and directly linked to the research question. A clear difference is observed between the groups in terms of reading abilities and executive functions, while ADHD symptoms remain similar. The negative correlations between reading performance and executive functions strengthen the validity of the findings. Although the impact and novelty are not explicitly discussed, the data are solid, statistically controlled, and provide a clear foundation for future replications. The conclusions are well grounded and limited to the results obtained, demonstrating a rigorous and responsible approach.